# Gaussian-Based Instance-Adaptive Intensity Modeling for Point-Supervised Facial Expression Spotting

**Yicheng Deng, Hideaki Hayashi, Hajime Nagahara**
Osaka University
`yicheng@is.ids.osaka-u.ac.jp, {hayashi, nagahara}@ids.osaka-u.ac.jp`

## Abstract

Point-supervised facial expression spotting (P-FES) aims to localize facial expression instances in untrimmed videos, requiring only a single timestamp label for each instance during training. To address label sparsity, hard pseudo-labeling is often employed to propagate point labels to unlabeled frames; however, this approach can lead to confusion when distinguishing between neutral and expression frames with various intensities, which can negatively impact model performance. In this paper, we propose a two-branch framework for P-FES that incorporates a Gaussian-based instance-adaptive Intensity Modeling (GIM) module for soft pseudo-labeling. GIM models the expression intensity distribution for each instance. Specifically, we detect the pseudo-apex frame around each point label, estimate the duration, and construct a Gaussian distribution for each expression instance. We then assign soft pseudo-labels to pseudo-expression frames as intensity values based on the Gaussian distribution. Additionally, we introduce an Intensity-Aware Contrastive (IAC) loss to enhance discriminative feature learning and suppress neutral noise by contrasting neutral frames with expression frames of various intensities. Extensive experiments on the SAMM-LV and CAS(ME)$^2$ datasets demonstrate the effectiveness of our proposed framework. Code is available at `https://github.com/KinopioIsAllIn/GIM`.

## 1 Introduction

Facial expressions play an important role in conveying human emotions as a typical form of nonverbal communication. Facial expressions can be divided into macro-expressions (MaEs) and micro-expressions (MEs). Macro-expressions are of high intensity, and they usually last between 0.5 and 4.0 seconds (Ekman, 2003a). Macro-expression analysis is important in various applications such as social robots (Rawal & Stock-Homburg, 2022), virtual reality (Ortmann et al., 2023), and so on. In contrast, micro-expressions are subtle and rapid (shorter than 0.5 seconds) (Ben et al., 2021). They are also used in many emotion-related applications, such as lie detection (Ekman & Friesen, 1969) and psychological counseling (Ekman, 2003b) since they are spontaneous and represent real emotions. Therefore, both macro- and micro-expression analysis are significant in human life.

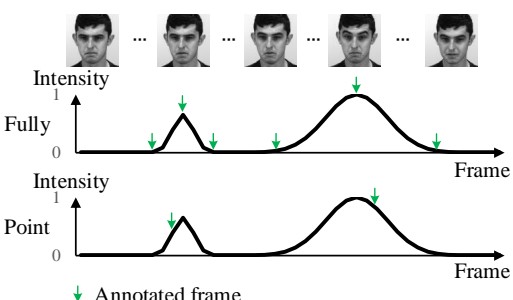

Figure 1: Different methods for annotation. The fully-supervised method requires annotating the onset, apex, and offset frames of each instance, whereas the point-supervised method requires annotating only a single frame for each instance.

Facial expression spotting (FES) is an important task in facial expression analysis. As a preliminary step to recognizing the specific emotional types of facial expressions, FES aims to localize facial expression instances in untrimmed videos, determining the onset and offset frames and classifying

the expression type (i.e., MaE or ME) for each instance. FES is crucial for accurately identifying various expressions in videos, enabling more precise emotion recognition and enhancing applications in human-computer interaction. Previous works (Yin et al., 2023; Yu et al., 2023; Deng et al., 2024a) have mainly focused on fully-supervised FES (F-FES). They usually extract optical flow features and develop deep learning models to analyze the extracted features, achieving good performance. The remarkable progress of F-FES can be attributed to the use of frame-level annotations.

To incorporate the findings of these F-FES studies into more practical problems under limited annotation cost, this paper investigates point-supervised FES (P-FES). As illustrated in Figure 1, in contrast to F-FES, which requires annotating the onset and offset frames with low expression intensity, P-FES requires only a single timestamp annotation at any intensity for each instance. This approach can significantly reduce the annotation burden and time required for training models, making it more feasible to deploy in real-world applications.

The potential challenge in P-FES lies in the difficulty of detecting complete expression instances in the absence of boundary labels while also suppressing neutral noise, which refers to specific neutral frames that do not convey any significant emotions but can interfere with the spotting process. Even though there is a notable lack of research on P-FES, many efforts have been devoted to point-supervised temporal action localization (P-TAL), which shares the same problem setting as our task, with the only difference being in target domains. Previous P-TAL methods (Ma et al., 2020; Lee & Byun, 2021; Zhang et al., 2024) typically employ a two-branch framework for class-agnostic score estimation and action classification. Subsequently, they mine reliable pseudo-action frames based on class-agnostic scores and feature similarity, then assign hard pseudo-labels to them. However, such a hard pseudo-labeling strategy may not be directly ap-

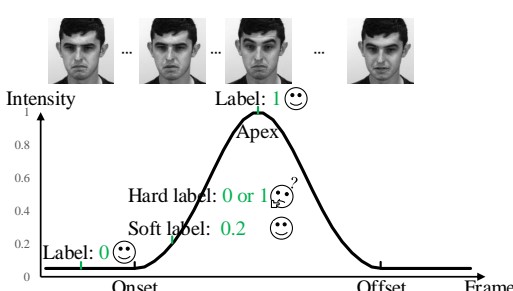

Figure 2: Motivation illustration. Due to the fact that expression frames have various intensities, it is difficult to describe this characteristic by hard pseudo-labeling. We use soft pseudo-labeling to learn the intensity distribution of each instance, reducing ambiguity in distinguishing between neutral and expression frames with various intensities.

plicable to the P-FES task, as it fails to help the model distinguish between neutral and expression frames with various intensities. As illustrated in Figure 2, although low-intensity expression frames are similar to neutral frames in intensity, they should be assigned a label of 1 when using hard pseudo-labeling—just like the high-intensity frames near the apex frames. In this case, hard pseudo-labels cannot precisely describe the characteristics of expression intensity, resulting in inaccurate class-agnostic output scores.

To solve the above-mentioned problem, we propose a two-branch framework that converts the binary classification-based class-agnostic branch into a regression-based expression intensity branch. In this paper, we assume that the expression intensity in each instance follows an individual smooth Gaussian distribution instead of a Bernoulli distribution. Based on this assumption, we propose Gaussian-based instance-adaptive Intensity Modeling (GIM) for P-FES. Specifically, we first employ a two-branch framework to estimate expression intensity scores and action scores. We then detect the pseudo-apex frame around each labeled frame and estimate the rough duration for each expression instance. Subsequently, we build a Gaussian distribution for each expression instance individually. The mean of the Gaussian distribution is determined by the feature of the pseudo-apex frame, and the variance is calculated by measuring the distance between the mean and the features of other pseudo-expression frames in the duration. Finally, we assign soft pseudo-labels as the expression intensity values for supervision and optimize the expression intensity branch. In addition, we introduce an Intensity-Aware Contrastive (IAC) loss on reliable pseudo-labeled frames from different classes, enhancing the model's ability to distinguish between neutral frames and expression frames with various intensities to suppress the influence of neutral noise and highlight expression frames.

Our contributions are as follows:

- We analyze the limitations of directly applying current P-TAL frameworks to P-FES and find that hard pseudo-labeling makes distinguishing between neutral and expression frames

with various intensities ambiguous. Thus, we propose a two-branch framework consisting of a regression branch to model facial expression intensity distribution using a soft pseudo-labeling strategy to reduce this ambiguity.

- We propose a Gaussian-based instance-adaptive intensity modeling module to model the expression intensity distribution of each expression proposal and assign soft pseudo-labels to pseudo-expression frames for supervision. In GIM, not only is soft pseudo-labeling crucial for P-FES, but apex frame detection is also significant for further expression recognition.

- We introduce intensity-aware contrastive learning on pseudo-labeled frames from different classes with various intensities to enhance the discriminative feature learning and suppress neutral noise.

## 2 RELATED WORKS

### 2.1 FACIAL EXPRESSION SPOTTING

Previous FES methods can be grouped into traditional methods and deep-learning methods. Traditional methods generally extract optical flow and analyze the pattern of each region of interest. Yuhong (2021) used the optical flow for facial alignment to eliminate the influence of head movement. Zhao et al. (2022) refined feature extraction and employed a Bayesian optimization algorithm for analyzing optical flow patterns. Wang et al. (2024) proposed skip-k-frame block-wise main directional mean optical flow (Liu et al., 2015) features and analyzed the M-pattern of these features.

Recently, many researchers have developed deep learning-based frameworks to solve the F-FES task. Leng et al. (2022) extended BSN (Lin et al., 2018), which was originally designed for TAL, and adapted it for FES. Yin et al. (2023) refined (Leng et al., 2022) approach by introducing graph convolutional networks and action unit (AU) label information. Yu et al. (2021; 2023) designed a two-branch framework based on A2Net (Yang et al., 2020) and introduced additional attention modules for facial expression spotting. Deng et al. (2024a) proposed an SW-MRO feature and introduced SpoT-GCN to improve the classification of individual frames. They then enhanced the framework by introducing SpotFormer (Deng et al., 2024b) and explored various model architectures.

### 2.2 POINT-SUPERVISED TEMPORAL ACTION LOCALIZATION

Many researchers have devoted their efforts to P-TAL to mitigate the intensive labor required for frame-level labels in fully-supervised TAL. Ma et al. (2020) proposed SF-Net to mine neighboring pseudo-action frames around each labeled frame to train the classifiers. Lee & Byun (2021) proposed to search for the optimal sequence for completeness learning using point labels. Fu et al. (2022) measured the confidence of each frame based on the feature similarity and rectified the output scores to assign reliable pseudo-labels. Zhang et al. (2024) proposed a two-stage framework to propagate high-confidence cues in point annotations at both snippet and instance levels. Xia et al. (2024) claimed that the most salient frame tends to appear in the central region of each instance and presented a proposal-level plug-in framework to relearn the aligned confidence of proposals to refine them.

Although P-FES, the focus of this study, has many similarities with P-TAL, research specifically dedicated to P-FES is extremely limited. To our knowledge, (Yu et al., 2024) is the only paper that explored P-FES, employing a framework similar to general P-TAL methods. In this paper, instead of employing hard pseudo-labeling, which may increase the ambiguity in distinguishing between neutral and expression frames with various intensities, we propose a Gaussian-based soft pseudo-labeling strategy to model the expression intensity distribution for each instance.

### 2.3 SOFT PSEUDO-LABELING

Soft pseudo-labeling is an advanced semi-supervised learning technique generally investigated in classification tasks. Unlike hard pseudo-labeling, which assigns a single class label to unlabeled data based on the model's most confident prediction, soft pseudo-labeling generates soft labels representing the full distribution of class probabilities, considering uncertainty in predictions. Nassar et al. (2023) proposed PROTOCON, which refines soft pseudo-labeling by knowledge of neighbors in a prototypical embedding space for semi-supervised image classification. Lukov et al. (2022)

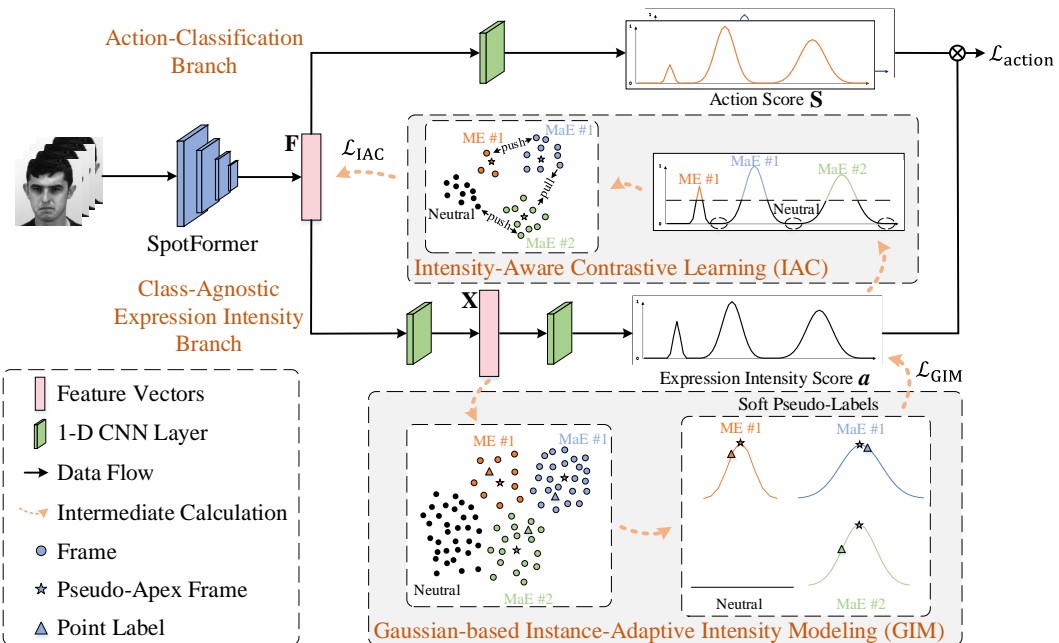

Figure 3: Overview of the proposed framework. The framework initially calculates the optical flow and extracts snippet features. These features are fed into a two-branch framework to obtain action and expression intensity scores. A GIM module is employed to build the Gaussian distribution for each expression instance and assign soft pseudo-labels to model the intensity distribution. An IAC module is employed to build contrasts among pseudo-labeled frames with various intensities to enhance feature learning and suppress neutral noise.

proposed to smooth multiple high-confidence classes in the logits based on their confidence while assigning a fixed low probability to low-confidence classes to handle noisy labels in in-the-wild facial expression recognition. Recently, several methods (Liang et al., 2022; Wu et al., 2023; Shen et al., 2024) built Gaussian Mixture Models to model class-wise feature distribution for semantic segmentation. Inspired by these works, we propose to construct an individual Gaussian distribution for each expression instance to assign soft pseudo-labels as direct intensity supervision signals and train a regression model to learn the expression intensity distribution, rather than the distribution of class probabilities. To the best of our knowledge, we are the first to investigate the application of soft pseudo-labeling for P-FES, providing a novel perspective on modeling expression intensity.

## 3 METHODOLOGY

### 3.1 PROBLEM FORMULATION

Given an untrimmed facial video $V = (v_i)_{i=1}^T$, we only have single timestamp annotation for each facial expression instance, i.e., $Y = (p_i, \boldsymbol{y}_i)_{i=1}^N$, where $p_i$ represents the annotated frame of the $i$-th expression instance, $N$ denotes the total number of ground-truth expression instances, and $\boldsymbol{y}_i$ denotes the multi-hot vector representing the action class (i.e., MaE and ME), respectively. Our objective is to detect as many expression instances as possible, localizing the boundary frames and determining the expression type for each instance.

### 3.2 BASELINE FRAMEWORK

The input video $V$ is first divided into $T$ overlapping snippets, where each snippet represents a short sequence of consecutive video frames that contain the temporal context of a single frame, following (Deng et al., 2024b). Then, we employ SpotFormer (Deng et al., 2024b) as the feature extractor to extract and embed optical flow features into feature vectors and concatenate them along the channel dimension, resulting in $\mathbf{F} \in \mathbb{R}^{T \times D}$, where $D$ denotes the dimension of each snippet feature. Then,

similar to general P-TAL works, we input the embedded feature vectors into a two-branch framework to estimate the class-agnostic expression intensity scores $\boldsymbol{a} \in \mathbb{R}^T$ and the action scores $\mathbf{S} \in \mathbb{R}^{T \times C}$, where $C$ represents the number of expression classes (i.e., MaE and ME). Note that the FES task does not involve emotion recognition; therefore, the estimated intensity scores and action scores are independent of emotional categories.

### 3.3 MOTIVATION

Due to the sparsity of point labels, propagating labels from labeled to unlabeled frames is key to enhancing model training and performance. Current P-TAL methods (Lee & Byun, 2021; Zhang et al., 2024) usually assign hard pseudo-labels to neighboring frames, frames with high class-agnostic scores, or frames that have high feature similarity with labeled frames. Then, the hard pseudo-labels are used to train a binary classification model for the class-agnostic branch and a multiclass classification model for the action classification branch. However, we observe that the hard pseudo-labeling strategy will cause ambiguity in FES when distinguishing between neutral and expression frames with various intensities. For example, expression frames near boundary frames have low expression intensity, which makes them have higher feature similarity with neutral frames than apex frames. Therefore, it is difficult to assign hard pseudo-labels to these low-intensity expression frames, resulting in the binary class-agnostic branch and hard pseudo-labeling being unsuitable for P-FES. To overcome this issue, we convert the binary classification-based class-agnostic branch into a regression-based expression intensity branch and propose GIM to assign soft pseudo-labels to frames with various intensities, modeling the expression intensity distribution of each instance.

### 3.4 GAUSSIAN-BASED INSTANCE-ADAPTIVE INTENSITY MODELING (GIM)

Our solution is based on the assumption that the expression intensity within each expression instance follows a smooth Gaussian distribution, with the apex frame corresponding to the peak intensity, which decreases symmetrically on both sides. Neutral frames are assumed to have an intensity of 0.

Figure 3 shows the proposed framework. Our framework consists of an expression intensity branch and an action classification branch. To model the intensity distribution for each expression instance, we build the instance-adaptive Gaussian distributions based on the intermediate feature representations $\mathbf{X} \in \mathbb{R}^{T \times D}$ from the expression intensity branch and the output intensity scores $\boldsymbol{a}$.

The algorithm for constructing the instance-adaptive Gaussian distributions and assigning soft pseudo-labels is described as follows.

**Step 1.** Given the output intensity scores $\boldsymbol{a}$ and a point label $p_i$ with expression class $c$, we detect the pseudo-apex frame $v_i^{\text{apex}}$ with the highest intensity score in a pre-defined range $I_i$:

$$v_i^{\text{apex}} = \arg\max_{j \in I_i} a_j, \tag{1}$$

where $I_i = \{n \in \mathbb{Z} \mid p_i - \frac{k_c}{4} \leq n \leq p_i + \frac{k_c}{4}\}$, and $k_c$ denotes the general duration of the $c$-th class expression instance (i.e., MaE or ME). The intermediate feature of the pseudo-apex frame, $\boldsymbol{x}_i^{\text{apex}}$, is selected as the $\boldsymbol{\mu}_i$ for the $i$-th Gaussian distribution $g_i$:

$$\boldsymbol{\mu}_i = \boldsymbol{x}_i^{\text{apex}}. \tag{2}$$

This selection strategy ensures that the pseudo-apex frame is the center of the Gaussian distribution and has the highest soft pseudo-label.

**Step 2.** Centered at the pseudo-apex frame $v_i^{\text{apex}}$, we estimate the rough duration $L_i$ for the $i$-th expression instance by filtering out the neighboring expression frames whose intensity score is larger than a threshold $\theta$:

$$L_i = |\{j \in J_i \mid a_j > \theta\}|, \tag{3}$$

where $J_i = \{n \in \mathbb{Z} \mid v_i^{\text{apex}} - \frac{k_c}{2} \leq n \leq v_i^{\text{apex}} + \frac{k_c}{2}\}$, and $a_j$ corresponds to the intensity score of $v_j$. Then, we expand the rough duration by a coefficient $\delta$ to consider unreliable low-intensity expression frames to complete the expression proposal. In each expression proposal, we measure the feature distance between each frame and the pseudo-apex frame; then, we calculate the variance $\sigma_i$ for the Gaussian distribution. The formulation can be described as:

$$\sigma_i = \sqrt{\frac{1}{\delta L_i} \sum_{j \in K_i} \|\boldsymbol{x}_j - \boldsymbol{\mu}_i\|_2^2}, \tag{4}$$

where $\| \cdot \|_2$ denotes the Euclidean distance, and $K_i = \{n \in \mathbb{Z} \mid v_i^{\text{apex}} - \frac{\delta L_i}{2} \le n \le v_i^{\text{apex}} + \frac{\delta L_i}{2}\}$.

**Step 3.** Finally, we build an unnormalized Gaussian distribution $g_i$ for the $i$-th expression instance:

$$g_i(\boldsymbol{x}_j; \boldsymbol{\mu}_i, \sigma_i) = \exp\left(-\frac{\|\boldsymbol{x}_j - \boldsymbol{\mu}_i\|_2^2}{2\sigma_i^2}\right), \quad j \in K_i. \tag{5}$$

Given the Gaussian distribution $g_i$, we can assign a soft pseudo-label to each pseudo-expression frame in the range of $(0, 1]$.

### 3.5 INTENSITY-AWARE CONTRASTIVE LEARNING ON PSEUDO-LABELED FRAMES

To further suppress neutral noise, highlight expression frames, and learn inter-class differences for action classification, we introduce contrastive learning (Khosla et al., 2020) to pseudo-labeled frames. Additionally, we consider the impact of intensity differences and propose an Intensity-Aware Contrastive (IAC) loss. The intuition is that the intensity differences between frames are independent of the class, and we should consider these intensity differences when building contrasts on pseudo-labeled frames. Specifically, for two samples with the same pseudo-class label, we focus less on pulling them together when their intensity difference is large; for two samples with different pseudo-class labels, we focus less on pushing them apart when their intensity difference is small.

To introduce contrastive learning, identifying the neutral frames is necessary since neutral labels are not provided. In the previous section, we focused on assigning soft pseudo-labels to pseudo-expression frames. Suppose we assign pseudo-expression labels to $N_{\text{exp}}$ frames. We employ the top-$k$ strategy to select pseudo-neutral frames from those not given pseudo-labels with the top-$k$ lowest expression intensity scores. The number of pseudo-neutral frames $N_{\text{neut}}$ is determined by:

$$N_{\text{neut}} = \min(N_{\text{exp}}, T - N_{\text{exp}}). \tag{6}$$

Then, we select reliable pseudo-expression frames with pseudo-intensity labels larger than 0.5 and build intensity-aware contrasts among reliable pseudo-neutral and pseudo-expression frames. Let $\mathcal{I}$ represent the set of reliable pseudo-expression and pseudo-neutral frames, and the loss function is formulated as:

$$\mathcal{L}_{\text{IAC}} = \sum_{i \in \mathcal{I}} \frac{-1}{|Q(i)|} \sum_{q \in Q(i)} \log \frac{w_{i,q} \exp(\boldsymbol{f}_i^\top \boldsymbol{f}_q / \tau)}{\sum_{e \in E(i)} w_{i,e} \exp(\boldsymbol{f}_i^\top \boldsymbol{f}_e / \tau)}, \tag{7}$$

$$w_{i,j} = \begin{cases} 1 - |\hat{a}_i - \hat{a}_j|, & \text{if } \widetilde{y}_i = \widetilde{y}_j \\ |\hat{a}_i - \hat{a}_j|, & \text{if } \widetilde{y}_i \ne \widetilde{y}_j \end{cases}, \tag{8}$$

where $E(i) := \mathcal{I} \backslash i$, and $Q(i) := \{q \in E(i) \mid \widetilde{y}_q = \widetilde{y}_i\}$ represents the set of samples in the video who has the same pseudo-class label with the $i$-th sample, $\boldsymbol{f}_i$ is the embedded feature of the $i$-th sample (the $i$-th snippet feature of $\mathbf{F} \in \mathbb{R}^{T \times D}$), $\tau \in \mathbb{R}^+$ is a scalar temperature parameter, $\hat{a}_i$ represents the pseudo-intensity label of the $i$-th sample, respectively.

### 3.6 TRAINING AND INFERENCE

#### 3.6.1 LOSS FUNCTION

General P-TAL methods (Lee & Byun, 2021; Zhang et al., 2024) treat the class-agnostic branch as a binary-classification branch and employ the binary cross-entropy (BCE) loss function to optimize the branch. In this paper, since we assume the expression intensity score in each expression instance follows a smooth Gaussian distribution instead of a Bernoulli distribution, we treat it as a regression task and employ the mean squared error (MSE) loss to optimize the expression intensity branch:

$$\mathcal{L}_{\text{GIM}} = \frac{1}{N_{\text{neut}} + N_{\text{exp}}} \sum_{i \in \mathcal{T}} (a_i - \hat{a}_i)^2, \tag{9}$$

where $\mathcal{T}$ represents the set of all pseudo-labeled frames, and $a_i$ and $\hat{a}_i$ represent the output intensity score and the corresponding soft pseudo-label of the frame $v_i$, respectively.

Following the previous work (Hong et al., 2021), due to the sparsity of expressions in the video, we add an L1 normalization loss on the intensity scores:

$$\mathcal{L}_{\text{norm}} = \|\boldsymbol{a}\|_1, \tag{10}$$

where $\| \cdot \|_1$ is a L1-norm function.

We also employ a video smooth loss to encourage temporal consistency in video output by ensuring that consecutive frames have similar predictions, stabling the training process:

$$\mathcal{L}_{\text{smooth}} = \frac{1}{T-1} \sum_{t=1}^{T-1} \|a_{t+1} - a_t\|_1. \tag{11}$$

For the action classification branch, we employ general cross-entropy loss. Due to the normalization loss and the tendency for most frames to have lower scores (neutral and low-intensity expression frames), the model tends to produce low expression intensity scores. Therefore, we first refine action scores by combining them with the intensity scores:

$$\bar{s}_{i,c} = s_{i,c} \cdot a_i, \quad i \in \{1, ..., T\}, c \in \{1, ..., C\}, \tag{12}$$

where $s_{i,c}$ denotes the original output probability that the $i$-th sample belongs to the $c$-th class expression. Then we calculate the action classification loss to encourage reliable expression frames to generate higher scores and reliable neutral frames to generate lower scores:

$$\mathcal{L}_{\text{action}} = -\frac{1}{|\mathcal{I}^+|} \sum_{i \in \mathcal{I}^+} \sum_{c=1}^{C} \widetilde{y}_{i,c} \log \bar{s}_{i,c} - \frac{1}{C|\mathcal{I}^-|} \sum_{i \in \mathcal{I}^-} \sum_{c=1}^{C} \log(1 - \bar{s}_{i,c}), \tag{13}$$

where $\mathcal{I}^+$ and $\mathcal{I}^-$ represent the set of reliable pseudo-expression frames and pseudo-neutral frames, and $\widetilde{y}_{i,c}$ represents the pseudo-class label of the $i$-th sample, respectively.

Finally, the total loss function can be summarized as:

$$\mathcal{L} = \mathcal{L}_{\text{GIM}} + \mathcal{L}_{\text{action}} + \lambda_1 \mathcal{L}_{\text{smooth}} + \lambda_2 \mathcal{L}_{\text{norm}} + \lambda_3 \mathcal{L}_{\text{IAC}}, \tag{14}$$

where $\lambda_1$, $\lambda_2$, and $\lambda_3$ are hyper-parameters for balancing the losses, which are determined empirically.

### 3.6.2 TRAINING PIPELINE

In the early training epochs, the output expression intensity scores are not sure to represent the expression intensity, and the highest score does not definitely represent the apex frame. Therefore, we employ an easy-to-hard learning paradigm and set several warm-up training epochs to make sure that the output intensity score can represent expression intensity. Specifically, 1) in the first stage, we assign hard pseudo-labels to adjacent frames around each labeled frame $p_i$ in range $[p_i - k_{s1}, p_i + k_{s1}]$. 2) In the second stage, we build the Gaussian distribution centered at the labeled frame and assign soft pseudo-labels in a pre-defined small range $p_i$ in range $[p_i - k_{s2}, p_i + k_{s2}]$. 3) In the third stage, we employ our proposed GIM module for soft pseudo-labeling and model training. The pseudo-apex frame and the range for soft pseudo-labeling are updated at each training epoch to enhance intensity-related feature learning.

### 3.6.3 INFERENCE

In the inference phase, we first obtain the expression intensity scores $a$ and suppressed action scores $\bar{\mathbf{S}}$. We then generate candidate expression proposals by using multiple thresholds for $a$, where each proposal includes consecutive frames with intensity scores higher than a given threshold. Each proposal is represented as $(s_i, e_i, c_i, p_i^{\text{OIC}})$, where $s_i, e_i, c_i$, and $p_i^{\text{OIC}}$ represent the onset frame, offset frame, expression type, and the outer-inner-contrastive (OIC) score (Shou et al., 2018), respectively. Specifically, $c_i$ is determined by applying a threshold of 0.5 to the action score of the pseudo-apex frame since it is the most representative frame for each expression instance. The OIC score can be calculated as follows:

$$p_i^{\text{OIC}} = \frac{1}{L_i} \sum_{t=s_i}^{e_i} a_t - \frac{2}{L_i} \left( \sum_{t=s_i - \frac{L_i}{4}}^{s_i - 1} a_t + \sum_{t=e_i+1}^{e_i + \frac{L_i}{4}} a_t \right), \tag{15}$$

$$L_i = e_i - s_i + 1. \tag{16}$$

Finally, we apply class-wise NMS (Bodla et al., 2017; Yin et al., 2023) to remove redundant ones.

Table 1: Comparison with the state-of-the-art methods on SAMM-LV and CAS(ME)$^2$ in terms of F1 score. The dagger †denotes the method originally for P-TAL and reproduced to P-FES.

| | Methods | SAMM-LV | | | CAS(ME)$^2$ | | |
|---|---|---|---|---|---|---|---|
| | | MaE | ME | Overall | MaE | ME | Overall |
| F-FES | SOFTNet (Liong et al., 2021) | 0.2169 | 0.1520 | 0.1881 | 0.2410 | 0.1173 | 0.2022 |
| | Concat-CNN (Yang et al., 2021) | 0.3553 | 0.1155 | 0.2736 | 0.2505 | 0.0153 | 0.2019 |
| | LSSNet (Yu et al., 2021) | 0.2810 | 0.1310 | 0.2380 | 0.3770 | 0.0420 | 0.3250 |
| | 3D-CNN (Yap et al., 2022) | 0.1595 | 0.0466 | 0.1084 | 0.2145 | 0.0714 | 0.1675 |
| | MTSN (Liong et al., 2022) | 0.3459 | 0.0878 | 0.2867 | 0.4104 | 0.0808 | 0.3620 |
| | ABPN (Leng et al., 2022) | 0.3349 | 0.1689 | 0.2908 | 0.3357 | 0.1590 | 0.3117 |
| | AUW-GCN (Yin et al., 2023) | 0.4293 | 0.1984 | 0.3728 | 0.4235 | 0.1538 | 0.3834 |
| | SpoT-GCN (Deng et al., 2024a) | 0.4631 | 0.4035 | 0.4454 | 0.4340 | 0.2637 | 0.4154 |
| | SpotFormer (Deng et al., 2024b) | 0.4447 | 0.4281 | 0.4401 | 0.5061 | 0.2817 | 0.4841 |
| P-FES | LAC(Lee & Byun, 2021)† | 0.3714 | 0.1983 | 0.3223 | 0.3889 | 0.0833 | 0.3598 |
| | HR-Pro(Zhang et al., 2024)† | 0.3395 | 0.1667 | 0.2895 | 0.3515 | **0.1345** | 0.3261 |
| | TSP-Net(Xia et al., 2024)† | 0.3152 | 0.1567 | 0.2703 | 0.3781 | 0.0571 | 0.3358 |
| | **Ours** | **0.4189** | **0.2033** | **0.3587** | **0.4395** | 0.0588 | **0.4000** |

Table 2: Ablation study on loss functions.

| Loss functions | | | | SAMM-LV | | | CAS(ME)$^2$ | | |
|---|---|---|---|---|---|---|---|---|---|
| $\mathcal{L}_{\text{GIM}}$(MSE) | $\mathcal{L}_{\text{GIM}}$(BCE) | $\mathcal{L}_{\text{IAC}}$ | Others | MaE | ME | Overall | MaE | ME | Overall |
| | | ✓ | ✓ | 0.3994 | **0.2189** | 0.3477 | 0.4151 | 0.0556 | 0.3785 |
| | ✓ | ✓ | ✓ | 0.2892 | 0.1095 | 0.2375 | 0.3246 | 0.0282 | 0.2877 |
| ✓ | | | ✓ | 0.4050 | 0.1519 | 0.3360 | 0.4207 | **0.0870** | 0.3872 |
| ✓ | | ✓ | ✓ | **0.4189** | 0.2033 | **0.3587** | **0.4395** | 0.0588 | **0.4000** |

## 4 EXPERIMENTS

### 4.1 EXPERIMENTAL SETTINGS

**Datasets.** We follow the protocol of MEGC2021 and validate our method on two datasets: SAMM-LV (Yap et al., 2020) and CAS(ME)$^2$ (Qu et al., 2017). The SAMM-LV dataset has 147 annotated videos from 32 subjects with 200 fps, including 343 MaE clips and 159 ME clips. The CAS(ME)$^2$ dataset has 98 annotated videos from 22 subjects with 300 MaE clips and 57 ME clips, and the frame rate is 30 fps. Since the frame rates of both datasets are different, we downsample the frame rate of SAMM-LV seven times to align the frame rates.

**Evaluation metrics.** We employ a leave-one-subject-out cross-validation strategy in the experiments. An expression proposal is considered true positive (TP) if the Intersection over Union (IoU) between the expression proposal and a ground-truth expression instance satisfies:

$$\frac{W_{\text{Proposal}} \cap W_{\text{GroundTruth}}}{W_{\text{Proposal}} \cup W_{\text{GroundTruth}}} \geq \theta_{\text{IoU}}, \tag{17}$$

where $\theta_{\text{IoU}}$ is the IoU threshold, set to 0.5. We calculate the F1 score to evaluate the performance of our model and compare it with other methods.

**Training details.** We generate single-frame annotations using a Gaussian distribution centered on the ground-truth apex frame for each instance. The model is trained by the Adam optimizer (Kingma, 2015) on both datasets for 100 epochs with a learning rate of $2.0 \times 10^{-5}$ and a weight decay of 0.1. The coefficient $\delta$ for duration estimation is set to 1.2. For the multi-stage training, the epochs for each stage are 1, 4, and 95, respectively. $k_c$ is set to 16 for ME and 32 for MaE, respectively. $k_{s1}$ is set to 3 for MEs and 5 for MaEs, $k_{s2}$ is set to 2 for MEs and 4 for MaEs. We set the loss weight $\lambda_*$ to 0.1, 0.3, and $2.0 \times 10^{-5}$ for SAMM-LV, and to 0.1, 2.5, and $1.4 \times 10^{-4}$ for CAS(ME)$^2$, respectively. The threshold $\theta$ for estimating the rough duration of each expression proposal decreases linearly from 0.8 to 0.5 over 30 epochs and then remains at 0.5 until the end.

Table 3: Ablation study on pseudo-labeling strategies.

| Strategy | SAMM-LV | | | CAS(ME)$^2$ | | |
| --- | --- | --- | --- | --- | --- | --- |
| | MaE | ME | Overall | MaE | ME | Overall |
| Hard | 0.3994 | **0.2189** | 0.3477 | 0.4151 | 0.0556 | 0.3785 |
| Soft | 0.1496 | 0.1036 | 0.1335 | 0.2705 | 0.0286 | 0.2380 |
| Class-wise | 0.2551 | 0.1140 | 0.2049 | 0.3072 | 0.0000 | 0.2598 |
| Ours | **0.4189** | 0.2033 | **0.3587** | **0.4395** | **0.0588** | **0.4000** |

Figure 4: Pseudo-label results of four expression instances. The line graph with blue dots represents the soft pseudo-labels assigned by our model; the leftmost and rightmost blue dots indicate the estimated expression duration for pseudo-labeling, while the peak dot indicates the pseudo-apex frame.

## 4.2 COMPARISON WITH STATE-OF-THE-ART METHODS

We first compare the performance with state-of-the-art (SOTA) deep learning methods, and the results are shown in Table 1. Since there is no prior P-FES method, we reproduce and apply several SOTA P-TAL methods to the P-FES task. Note that for a fair comparison, we use the same feature extractor as in our method when reproducing these SOTA P-TAL methods, which can significantly improve the performance of P-FES. It can be seen that our method outperforms SOTA point-supervised methods by 11.3% on SAMM-LV and 11.2% on CAS(ME)$^2$. Note that the separate F1 scores for MaE and ME spotting in Table 1 represent their performance when we achieved the optimal overall performance, rather than their individual best performances. This partially explains why our ME spotting performance on CAS(ME)$^2$ is lower than that of other SOTA methods. We also show the results of several F-FES methods for comparison, and the results demonstrate that our method achieves competitive performance in MaE spotting but a relatively low ME spotting performance. The reason is that our method focuses on significantly suppressing neutral noise, which may overshadow extremely subtle micro-expressions without the help of precise frame-level annotations.

## 4.3 ABLATION STUDY

### 4.3.1 LOSS FUNCTION

We conduct ablation studies on loss functions to verify the effectiveness of our proposed modules. The results are shown in Table 2. By comparing the results of using the MSE loss and the BCE loss, we can see that treating the expression intensity branch as a regression task performs better than treating it as a binary classification task. This is because when our GIM generates soft labels for low-intensity expression frames, and these labels accurately describe the intensity, using the MSE loss allows them to be treated as direct intensity supervision without causing a large loss value. However, if we use the BCE loss, even when the soft pseudo-labels are accurate enough, the loss value could still be large, severely affecting model training. The first row of Table 2 shows that when we only use the proposed GIM to estimate the duration of each expression instance and assign hard pseudo-labels, the performance is lower. The results demonstrate that our proposed GIM module can accurately describe the intensity of each expression frame and improve the performance. In addition, the third row and the fourth row in Table 2 also demonstrate the effectiveness of our proposed IAC loss.

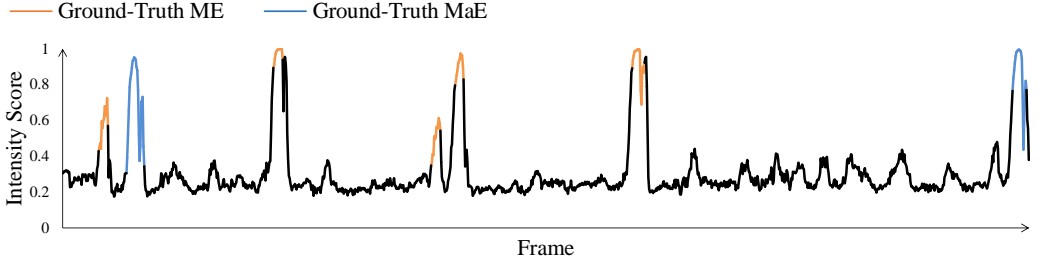

Figure 5: Expression intensity results of an entire video.

### 4.3.2 PSEUDO-LABELING STRATEGY

To verify the effectiveness of our proposed GIM, we conduct ablation studies on various pseudo-labeling strategies. In Table 3, **Hard** denotes our implemented hard pseudo-labeling method, which is the same as the first row in Table 2. **Soft** refers to using cosine similarity in the feature space as the soft pseudo-label instead of employing the proposed GIM, and **Class-wise** indicates that we use the class-wise average feature of point labels as the $\mu$ of the Gaussian distribution instead of detecting the pseudo-apex frame. The comparison with **Hard** and **Soft** demonstrates the effectiveness of our proposed GIM for soft pseudo-labeling. Additionally, the comparison with **Class-wise** highlights the effectiveness of our instance-adaptive approach and the choice of the $\mu$ for the Gaussian distribution. This is because when we select the class-wise average feature corresponding to point labels as the $\mu$ of the Gaussian distribution, even expressions within the same class can vary greatly in intensity. Therefore, considering the whole class instead of individual expression instances may cause some expressions to be assigned low soft labels, leading the model to ignore certain expressions, which negatively affects the performance. In summary, the results demonstrate the effectiveness of our instance-adaptive Gaussian distribution, built based on pseudo-apex frames and feature distances, for describing the expression intensity distribution.

### 4.4 QUALITATIVE EVALUATION

**Pseudo-label results.** For an intuitive illustration, we present some qualitative results of pseudo-labels, which are shown in Figure 4. According to the results, we can see that our GIM can detect apex frames more accurately than just using point labels as the pseudo-apex frames. In addition, our GIM can precisely estimate the duration of the expression instance, thus assigning reliable soft pseudo-labels.

**Intensity score results of an entire video.** We present the expression intensity score results for an entire video, as shown in Figure 5. The results demonstrate that our method significantly suppresses neutral noise while maintaining the intensity of expressions, highlighting the effectiveness of our proposed method.

## 5 CONCLUSION

In this paper, we investigated point-supervised facial expression spotting (P-FES). For this purpose, we proposed a two-branch framework by converting the general binary classification-based class-agnostic branch to a regression-based expression intensity branch to model the expression intensity distribution of each expression instance. In the expression intensity branch, we introduced a Gaussian-based instance-adaptive Intensity Modeling (GIM) module for soft pseudo-labeling. During training, we detected the pseudo-apex frame around each labeled frame and estimated the rough duration of each expression instance. Then, we built the Gaussian distribution centered at the pseudo-apex frame and assigned soft pseudo-labels to all potential expression frames in the estimated duration. In addition, we introduced an Intensity-Aware Contrastive (IAC) loss on pseudo-neutral frames and pseudo-expression frames with various intensities to enhance feature learning and further suppress neutral noise. Extensive quantitative and qualitative experiments on the SAMM-LV and CAS(ME)$^2$ datasets demonstrated the effectiveness of our proposed method.

ACKNOWLEDGMENTS

This work was partially supported by Innovation Platform for Society 5.0 from Japan Ministry of Education, Culture, Sports, Science and Technology, and JSPS KAKENHI Grant Number JP24K03010. This work was also supported by JST SPRING, Grant Number JPMJSP2138.

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

# A APPENDIX

## A.1 DISCUSSION ON CONSECUTIVE FACIAL EXPRESSION SPOTTING

In typical scenarios, facial expression spotting often focuses on individual expressions. However, two or more facial expressions may also occur consecutively within a short time, each with distinct apexes. This raises challenges in learning accurate intensity outputs and spotting consecutive expression instances.

To address this challenge, we categorize the issue into two cases: 1) consecutive expression instances of the same class (either MaEs or MEs); 2) consecutive expression instances of different classes.

**Soft pseudo-labeling.** For the first case, a frame might have multiple soft pseudo-labels when multiple close point-labels are used to build partially overlapping Gaussian distributions for soft pseudo-labeling. When assigning a soft pseudo-label to a single frame that already has a soft pseudo-label (not 0), we randomly retain one label and discard the other. This is done to prevent our model from being biased toward either high-intensity or low-intensity output, which could affect the performance of our model.

We further validate the effectiveness of our solution by comparing it with two other optional solutions: 1) keeping the higher soft label when two soft labels are generated for a single frame, denoted as 'Higher'; 2) keeping the lower soft label when two soft labels are generated for a single frame, denoted as 'Lower'. The results are shown in Table 4, which demonstrate that the 'random selection' strategy achieves the best overall performance.

Table 4: Ablation study on pseudo-labeling strategies for consecutive expression instances.

| | SAMM-LV | | | CAS(ME)$^2$ | | |
|---|---|---|---|---|---|---|
| Strategy | MaE | ME | Overall | MaE | ME | Overall |
| Higher | 0.4104 | 0.1885 | 0.3473 | 0.4338 | **0.0882** | 0.3988 |
| Lower | **0.4304** | 0.1579 | 0.3570 | 0.4276 | 0.0579 | 0.3891 |
| **Random (Ours)** | 0.4189 | **0.2033** | **0.3587** | **0.4395** | 0.0588 | **0.4000** |

For the second case, it is practical for us to use two arrays to store the soft pseudo-labels for the two different classes separately, even if the two classes share the same intensity output. This means that one frame may have two soft pseudo-labels for intensity supervision. When calculating the loss $\mathcal{L}_{\text{GIM}}$ between the output intensity and pseudo-labels, both soft pseudo-labels are used, and the two loss values are averaged. This strategy enables our model to learn various intensity information of possible composite expressions that occur in different facial areas (e.g., a person is performing an MaE with the eyebrows, and an ME occurs at the mouth corner later).

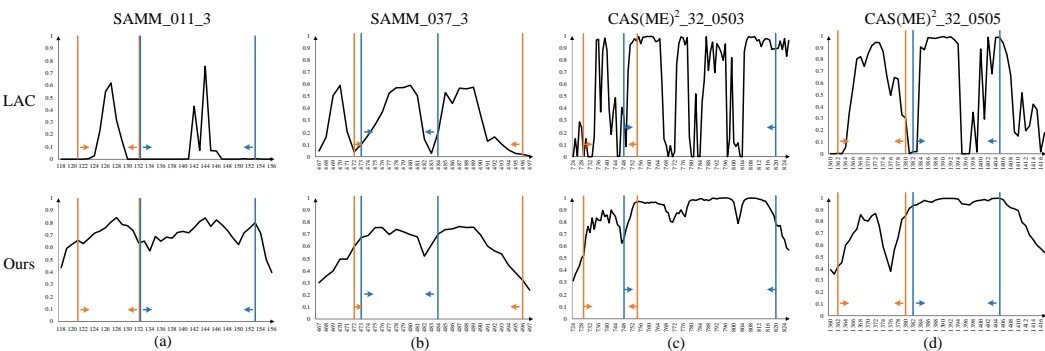

Figure 6: Expression intensity results for several examples of consecutive facial expressions. Each subfigure includes the expression intensity outputs of LAC (Lee & Byun, 2021) (first row) and our model (second row). Orange lines and blue lines represent the ground-truth boundaries of two consecutive expression instances, respectively.

**Inference phase.** During the post-processing, a trough will appear in the intensity output between two consecutive expressions, allowing us to use the multi-threshold strategy to detect and separate

them by the trough. The expression class is determined by each proposal's pseudo-apex frame, which has the highest intensity score.

Figure 6 presents more intuitive qualitative results. Combining with the multi-threshold strategy, our method can effectively detect and separate consecutive expression instances.

## A.2  HYPER-PARAMETER $k_c$ FOR MACRO-EXPRESSION

As the hyper-parameter of the range for soft pseudo-labeling, $k_c$ can affect the model performance significantly. In our method, $k_c$ is set based on the understanding that general MEs last less than 0.5 seconds, while MaEs last longer than 0.5 seconds. Since we pre-process the datasets to standardize the frame rate at 30 fps, 0.5 seconds corresponds to 15 frames. In practice, we set $k_c$ to 16 for MEs. However, due to the varying length of MaEs, the choice of $k_c$ for MaEs can affect the accuracy of assigned soft pseudo-labels. Small number is not helpful for assigning enough reliable soft pseudo-labels while a large number could result in assigning soft labels to noisy neutral frames or to expression frames of other expression instances. Therefore, we evaluated several choices of $k_c$ for MaEs, and the results are shown in Table 5.

Table 5: Ablation study on choices of $k_c$ for macro-expressions.

| $k_c$ for MaEs | SAMM-LV | | | CAS(ME)$^2$ | | |
|---|---|---|---|---|---|---|
| | MaE | ME | Overall | MaE | ME | Overall |
| 24 | **0.4240** | 0.1700 | 0.3541 | 0.4272 | 0.0317 | 0.3898 |
| **32** | 0.4189 | **0.2033** | **0.3587** | **0.4395** | 0.0588 | **0.4000** |
| 48 | 0.4144 | 0.1735 | 0.3510 | 0.4172 | **0.0822** | 0.3811 |
| 64 | 0.3833 | 0.1826 | 0.3271 | 0.4046 | 0.0563 | 0.3682 |

The results show that optimal performance is achieved when $k_c$ is set to 32 for MaEs, which is suitable for assigning enough reliable pseudo-MaE frames and enhancing feature learning.

## A.3  EVALUATION OF PSEUDO-APEX FRAME DETECTION

Although the facial expression spotting task only requires detecting the onset and offset frames, the apex frame is crucial for further emotion recognition. Therefore, we evaluate the performance of our model in detecting pseudo-apex frames. Figure 7 shows that as training proceeds, the average frame distance between pseudo-apex and ground-truth apex frames decreases and stabilizes at a low level, which demonstrates the effectiveness of our method in apex frame detection.

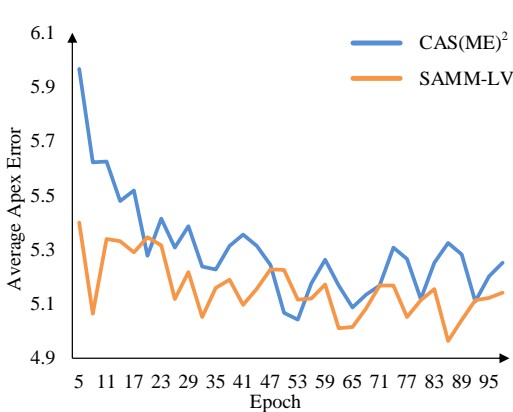

Figure 7: Changes in the average frame distance between pseudo-apex and ground-truth apex frames during training, evaluated starting from the 5th epoch.

