# OpenReview forum: "Gaussian-Based Instance-Adaptive Intensity Modeling for Point-Supervised Facial Expression Spotting"
_ICLR.cc/2025/Conference — ICLR 2025 Poster_

### Official Review · Reviewer_2FHD · 2024-11-02

**Soundness:** 3
**Presentation:** 3
**Contribution:** 3
**Rating:** 6
**Confidence:** 4

**Summary:**

The article proposes a two-branch framework for Point-supervised Facial Expression Spotting (P-FES), which includes a Gaussian-based Instance-Adaptive Intensity Modeling (GIM) module for soft pseudo-labeling. This framework aims to address the problem of locating facial expression instances in untrimmed videos, requiring only a timestamp label for each instance during training.

**Strengths:**

1、The motivation of this article is well-founded, and the proposed method effectively addresses the problem of point-supervised facial expression recognition.

2、The experiments are quite comprehensive, and the comparisons with other methods are reasonable.

**Weaknesses:**

Some details are unclear; see the "Questions" section below for more information.

**Questions:**

1、Not all facial expression instances conform to a Gaussian distribution in every situation, such as when someone is suddenly startled. The authors do not seem to address how to handle this special case.

2、I hope that Figure 4 can provide a clearer explanation.

---

> ### Author Response · Authors · 2024-11-19
> **Response to Reviewer 2FHD**
>
> **Q1. Not all facial expression instances conform to a Gaussian distribution in every situation, such as when someone is suddenly startled. The authors do not seem to address how to handle this special case.**
>
> **A1.** Thank you very much for your insightful comments.
> The Gaussian distribution we build is based on the intensity-related feature distance between each frame and the pseudo-apex frame, making it adaptive to each expression instance. For a given frame, if it is distant from the pseudo-apex frame in feature space, its intensity label will be lower, which is adaptive to the feature distance rather than the temporal distance. This instance-adaptive Gaussian distribution better describes the actual intensity changes.
>
> We believe a Gaussian distribution is suitable for modeling expression intensity because each expression instance transitions from onset -> apex -> offset. Even micro-expressions, such as the sudden startle, also occur over a certain period. Therefore, we investigate and find that Gaussian distribution based on intensity-related feature distances is effective for modeling expression intensity distributions.
> In Table 3, we also compare our method with other intensity modeling methods, such as ‘Soft’. In ‘Soft’, the soft pseudo-labels are directly calculated based on the feature similarity between pseudo-apex and expression frames, and the actual results are poor. This further demonstrates that our proposed GIM can accurately describe the expression intensity change in each expression instance.
>
> In Section 4.3.2 of the revised manuscript, we further summarized the results of the ablation studies and highlighted the effectiveness of our method for modeling instance adaptive expression intensity distribution even if individual expression instances may not strictly follow a standard Gaussian distribution. Please kindly refer to it.
>
> **Q2. I hope that Figure 4 can provide a clearer explanation.**
>
> **A2.** Thank you for pointing out this issue. Each subfigure contains information about a single expression instance: the ground-truth point-label (yellow triangle), the ground-truth temporal boundaries (dotted orange lines, after updating), the ground-truth apex frame (red star), and pseudo-labels generated by our model (dark blue dots on the black line graph). The two dotted orange lines in each subfigure represent the ground-truth boundaries of the expression instance, with the left line marking the onset frame and the right line marking the offset frame. The line graph with dots in each subfigure represents the soft pseudo-labels we assigned, indicating the estimated duration and the pseudo-apex frame (the peak of the line graph). For example, in Figure 4(a), the ground-truth onset frame is 1, the ground-truth offset frame is frame 13, our estimated duration is from frame 1 to frame 11, and the detected pseudo-apex frame is frame 6.
>
> We have updated Figure 4 to enhance its clarity in our revised manuscript, please kindly refer to it.

---

> ### Comment · Reviewer_2FHD · 2024-11-26
> **Final Rating**
>
> I want to thank the other reviewers and authors. I won't change the score. I am inclined to accept this paper.

---

> > ### Author Response · Authors · 2024-11-26
> > **Official Comment by Authors**
> >
> > We sincerely thank you for your positive evaluation.

---

### Official Review · Reviewer_qxKN · 2024-11-03

**Soundness:** 2
**Presentation:** 3
**Contribution:** 2
**Rating:** 6
**Confidence:** 5

**Summary:**

In this paper, the author introduced a two-branch framework for Point-supervised facial expression spotting (P-FES) task.
The proposed framework consists of two main branches: (1) Expression Intensity branch with a proposed Gaussian-based instance-adaptive Intensity Modeling (GIM) to model the intensity distribution of expression proposals; and (2) Action-Classification branch to classify the expression class from the proposals.
An Intensity-Aware Contrastive Learning on pseudo-labeled frame is also presented to enhance the discriminative feature learning process.
Experiments on SAMM-LV and CAS(ME)2 have shown the advantages of the proposed approach.

**Strengths:**

- The paper addresses interesting issue of facial expression spotting.
- Ablation study has shown some contributions of the proposed terms.

**Weaknesses:**

Although the soft-label is well-motivated, it has been widely used as a label smoothing technique in literature.
Moreover, the training process is complicated and ad-hoc with 3 training stages. However, the improvement is minor and cannot out-perform F-FES techniques.


1. During the training process, warm-up epochs are very important as both features and scores are not reliable for mining steps (i.e. extract pseudo-apex frame, gaussian modeling and reliable pseudo-neutral frames). Then, hard label technique is required. As a result, the proposed framework can only give a minor improvement on top of hard label technique.

2. In the proposed framework, one of the mining step is to extract pseudo-neutral frames. This will be challenging for ME classes as the micro-expression frames will be very similar to neutral frames and their feature difference is minor. How can the proposed framework extract reliable pseudo-neutral frames and achieve better performance on ME classes ?

3. How can the duration of each expression class be chosen?

4. More analysis on the pseudo-apex frame is recommended.
For example, How are the pseudo-apex frames changing during different training epochs?

**Questions:**

Please address the concerns in the Weaknesses section.

---

> ### Author Response · Authors · 2024-11-19
> **Response to Revewer qxKN**
>
> **Q1. Although the soft-label is well-motivated, it has been widely used as a label smoothing technique in literature.**
>
> **A1.** Thank you very much for appreciating our motivation and insights. Recently, soft pseudo-labeling has become a popular semi-supervised learning technique in various applications. However, our method is the first to apply soft pseudo-labeling into facial expression spotting. In addition, unlike general methods, we do not simply use soft pseudo-labeling to model the distribution of class probabilities for a general classification task. Instead, we adaptively model the facial expression intensity distribution for each expression instance as a direct intensity supervision signal. This strategy can be used to describe the intensity characteristics of each frame. It is implemented by training a regression model, which we have proved to be effective in the experimental section. Furthermore, the assignment of the soft pseudo-labels incorporates our several considerations: building the instance-adaptive Gaussian distribution, identifying the pseudo-apex frame as the $\mu$, estimating the range for soft pseudo-labeling, and considering intensity-related feature distance. Additionally, our proposed IAC loss also utilizes the soft pseudo-labels.
>
> We emphasized the difference between general soft pseudo-labeling application and our method in Section 2.3 in the revised manuscript. Please kindly refer to it.
>
>
> **Q2. Moreover, the training process is complicated and ad-hoc with 3 training stages. However, the improvement is minor and cannot out-perform F-FES techniques.**
>
> **A2.** Thank you very much for your comments. The training process described in section 3.6.2 is indeed detailed. We are sorry for describing it complicated. In the revised manuscript, we changed the description about this part by first summarizing the training process as an ‘easy-to-hard learning paradigm’. In the early training epochs, we assign confident pseudo-labels based on point-labels and neighboring frames within a small range. After the warm-up epochs, we employ our proposed GIM module to assign soft pseudo-labels to more frames.
>
> The experimental results in Table 1 show that our proposed method outperforms SOTA point-supervised methods by 11.29% and 11.17% on SAMM-LV and CAS(ME)$^2$, respectively, and even outperforms most F-FES methods. While we may not outperform all F-FES methods, our method achieves a competitive performance with very limited annotations, which can significantly reduce the time and effort for annotation, making our method more feasible in real world applications. Further improving the performance of our P-FES method and reducing the performance gap between P-FES and F-FES methods will be one of our future works.

---

> ### Author Response · Authors · 2024-11-19
> **Response to Reviewer qxKN (part 2)**
>
> **Q3. During the training process, warm-up epochs are very important as both features and scores are not reliable for mining steps (i.e. extract pseudo-apex frame, gaussian modeling and reliable pseudo-neutral frames). Then, hard label technique is required. As a result, the proposed framework can only give a minor improvement on top of hard label technique.**
>
> **A3.** In the early warm-up epochs, we conservatively assign hard pseudo-labels around point-labels within a very small range to learn basic knowledge about expression intensity for only one epoch. Then, we gradually employ our proposed GIM to assign soft pseudo-labels to avoid causing ambiguity between neutral and expression frames with various intensities, suppressing neutral noises.
>
> The comparison with SOTA hard pseudo-labeling methods in Table 1 shows that our proposed method outperforms them by 11.29% and 11.17% on SAMM-LV and CAS(ME)$^2$, respectively.
> The ‘Hard’ presented in Table 3 is a part of our work, which are the same as the first row in Table 2, employing the proposed IAC loss and GIM for duration estimation and hard pseudo labeling. Introducing our soft pseudo-labeling strategy further improves performance by 3.2% and 5.7% on SAMM-LV and CAS(ME)$^2$, respectively. The comprehensive ablation studies show the effectiveness of each proposed module.
>
> In addition, although the facial expression spotting task only requires detecting the onset and offset frames, the apex frame is crucial for further emotion recognition. Therefore, we believe that our work on soft pseudo-labeling and pseudo-apex frame detection is not only important for improving P-FES performance but also significant for further expression recognition.
>
> In the revised manuscript, we clarified the implementation of 'Hard' in Table 3 in Section 4.3.2. And we emphasized the significance of detecting pseudo-apex frames in Appendix A.3.
>
> **Q4. In the proposed framework, one of the mining step is to extract pseudo-neutral frames. This will be challenging for ME classes as the micro-expression frames will be very similar to neutral frames and their feature difference is minor. How can the proposed framework extract reliable pseudo-neutral frames and achieve better performance on ME classes?**
>
> **A4.** The mining of reliable pseudo-labeled frames is attributed to the point-labels. In the pseudo-labeling process, we mine pseudo-expression frames first. Given the point-labels, even ME instances with very low intensity can be assigned pseudo-expression labels, and the pseudo-apex frame will be assigned a label of 1 to enhance the output scores of subtle ME frames. After assigning pseudo-labels to all pseudo-expression frames, we identify pseudo-neutral frames with the top-k lowest intensity scores. Please note that the mining of pseudo-neutral frames does not take pseudo-expression frames into account.
>
> Therefore, the extraction of pseudo-neutral and pseudo-expression frames are performed sequentially. Specifically, we first extract pseudo-expression frames based on the point-labels to make sure subtle expression frames can also be mined. Then, we extract pseudo-neutral frames with lowest intensity scores.
>
> In the revised manuscript, the line 288 - 292 in Section 3.5 mentioned the order of mining pseudo-expression and pseudo-neutral frames. Please refer to it.

---

> ### Author Response · Authors · 2024-11-19
> **Response to Reviewer qxKN (part 3)**
>
> **Q5. How can the duration of each expression class be chosen?**
>
> **A5.** Thank you for pointing out this question. In Section 3.4, detecting the pseudo-apex frame and building the Gaussian distribution involves a hyper-parameter $k_c$, which defines a range for assigning soft pseudo-labels. This hyper-parameter is set based on the understanding that general MEs last less than 0.5 seconds, while MaEs last longer than 0.5 seconds. Since we pre-process the datasets to standardize the frame rate at 30 fps, 0.5 seconds corresponds to 15 frames. In practice, we set $k_c$ to 16 for MEs and 32 for MaEs. Please note that we didn’t set $k_c$ for MaEs to a large number because we aim to assign reliable soft pseudo-labels that describe intensity for reliable expression frames. This selection also avoids severe data-imbalance between MaEs and MEs during training. Setting $k_c$ for MaEs to a very large number could result in assigning soft labels to noisy neutral frames or to expression frames of other expression instances.
>
> We evaluated several choices of $k_c$ for MaEs, and the results are shown below:
>
> |                 |          SAMM-LV          |   |         CAS(ME)$^2$         |
> |:---------------:|:----------------------:|---|:----------------------:|
> | $k_c$ (for MaE) |  MaE \| ME \| Overall  |   |  MaE \| ME \| Overall  |
> | 24              | 0.4240\|0.1700\|0.3541 |   | 0.4272\|0.0317\|0.3898 |
> | 32              | 0.4189\|0.2033\|0.3587 |   | 0.4395\|0.0588\|0.4000 |
> | 48              | 0.4144\|0.1735\|0.3510 |   | 0.4172\|0.0822\|0.3811 |
> | 64              | 0.3833\|0.1826\|0.3271 |   | 0.4046\|0.0563\|0.3682 |
>
> The results show that optimal performance is achieved when $k_c$ is set to 32 for MaEs, which is suitable for assigning enough reliable pseudo-MaE frames and enhancing feature learning.
>
> The above is explained in Appendix A.2 in detail. Please kindly refer to it.
>
> **Q6. More analysis on the pseudo-apex frame is recommended. For example, How are the pseudo-apex frames changing during different training epochs?**
>
> **A6.** Thank you very much for your valuable suggestion. We discussed about the importance of detecting apex-frames and evaluated the average error between the ground-truth apex frames and the pseudo-apex frames in each epoch on SAMM-LV and CAS(ME)$^2$.
> The discussion and the results are summarized in Appendix A.3. Please refer to it in our revised manuscript. The results show that during the training process, the apex frame error decreases and stabilizes at a low level, which demonstrates the effectiveness of our method in detecting the pseudo-apex frames.

---

> > ### Comment · Reviewer_qxKN · 2024-11-27
> > **Final Rating**
> >
> > I acknowledge the response from the authors.
> > Although I still have concerns on the novelty and efficiency of the proposed approach, the authors have addressed part of these concerns.
> > In general, the paper is in a complete shape with experiments to support the design. It can act as a baseline for approaches along this direction.
> > Therefore, I increase my rating.

---

> > > ### Author Response · Authors · 2024-11-27
> > >
> > > We sincerely thank you for your thorough reviews and for raising the rating score.

---

### Official Review · Reviewer_nGC4 · 2024-11-04

**Soundness:** 3
**Presentation:** 3
**Contribution:** 3
**Rating:** 6
**Confidence:** 4

**Summary:**

This paper treats facial expression spotting as a point-supervised temporal action localization (P-TAL) problem. Drawing from existing P-TAL techniques, the authors propose a two-branch framework for localizing facial expressions. One branch, known as the action-classification branch, predicts the category scores of the facial expression $S$, while the other branch predicts its intensity score $a$.

The framework utilizes a proposed Gaussian-based Instance-Adaptive Intensity Modeling approach to establish soft pseudo-labels for each frame based on the predicted intensity and the given point label. The expression intensity branch is optimized by minimizing the difference between the predicted intensity and the soft pseudo-label, while the action-classification branch employs cross-entropy loss, with class labels also derived from the soft pseudo-label. To reduce neutral noise further, the authors introduce an intensity-aware contrastive loss.

The proposed method is evaluated on two datasets containing both macro and micro facial expressions. Experimental results indicate that this approach surpasses state-of-the-art P-TAL methods in the task of facial expression spotting.

**Strengths:**

1. The setting of point-supervised facial expression spotting is more practical than fully-supervised facial expression spotting.

2. The proposed intensity-aware contrastive learning is interesting. Rather than simply pulling together the same-class instances and pushing apart the different-class instances, the authors consider the influences of intensity, i.e., less attention should be paid to the same-class instances with large intensity differences, and to the different-class instances with small intensity differences.

**Weaknesses:**

1. In the proposed method, the intensity score is shared across all facial expression classes. However, two or more facial expressions may occur within a short clip, each with distinct apexes. This raises a concern: what happens if the method is applied to an untrimmed video containing two closely timed apexes of different facial expressions? It would be beneficial to discuss how the model handles this scenario.

2. The presentation of the inference lacks sufficient detail. Specifically, it would be helpful to clarify what the outer-inner-contrastive score is and how these scores contribute to determining the temporal range and position of the facial expressions.

3.  The soft pseudo-label is established based on the predicted intensity scores. However, these predicted intensity scores can fluctuate throughout the training process. It would be helpful to clarify how the soft pseudo-label is updated in response to these changing intensity scores.

4. The legend of Fig 4 is unclear. The marks of ground-truth boundary frame are missing.

**Questions:**

Please refer the the questions.

---

> ### Author Response · Authors · 2024-11-19
> **Response to Reviewer nGC4**
>
> Thank you very much for the positive evaluation.
>
> **Q1. In the proposed method, the intensity score is shared across all facial expression classes. However, two or more facial expressions may occur within a short clip, each with distinct apexes. This raises a concern: what happens if the method is applied to an untrimmed video containing two closely timed apexes of different facial expressions? It would be beneficial to discuss how the model handles this scenario.**
>
> **A1.** Thank you for your insightful comment. This type of scenario does exist in the datasets. We would like to introduce our solution for this issue, which has already been implemented in our original method.
>
> The problem can be categorized into two cases: 1) consecutive expression instances of the same class (either MaEs or MEs); 2) consecutive expression instances of different classes.
>
> Training phase: For the first case, a frame might have multiple soft pseudo-labels when multiple close point-labels are used to build partially overlapping Gaussian distributions for soft pseudo-labeling. When assigning a soft pseudo-label to a single frame that already has a soft pseudo-label (not 0), we randomly retain one label and discard the other. This is done to prevent our model from being biased toward either high-intensity or low-intensity output, which could affect the performance of our model.
>
> We further validated the effectiveness of our solution by comparing it with two other optional solutions: 1) keeping the higher soft label when two soft labels are generated for a single frame, denoted as ‘Higher’; 2) keeping the lower soft label when two soft labels are generated for a single frame, denoted as ‘Lower’. The results, shown below, demonstrate that the ‘random selection’ strategy achieves the best performance.
>
> |               |          SAMM-LV          |   |         CAS(ME)$^2$         |
> |:-------------:|:----------------------:|---|:----------------------:|
> |               |  MaE \| ME \| Overall  |   |  MaE \| ME \| Overall  |
> | Higher        | 0.4104\|0.1885\|0.3473 |   | 0.4338\|0.0882\|0.3988 |
> | Lower         | 0.4304\|0.1579\|0.3570 |   | 0.4276\|0.0579\|0.3891 |
> | Random (Ours) | 0.4189\|0.2033\|0.3587 |   | 0.4395\|0.0588\|0.4000 |
>
> For the second case, it is practical for us to use two arrays to store the soft pseudo-labels for the two different classes separately, even if the two classes share the same intensity output. This means that one frame may have two soft pseudo-labels for intensity supervision. When calculating the loss $\mathcal{L}_{\mathrm{GIM}}$ between the output intensity and pseudo-labels, both soft pseudo-labels are used, and the two loss values are averaged. This strategy enables our model to learn various intensity information of possible composite expressions that occur in different facial areas (e.g., a person is performing an MaE with the eyebrows, and an ME occurs at the mouth corner later).
>
> During the inference phase, a trough will appear in the intensity output between two consecutive expressions, allowing us to use the multi-threshold strategy to detect and separate them by the trough. The expression class is determined by each proposal's pseudo-apex frame, which has the highest intensity score.
>
> The above is explained in Appendix A.1 in detail. Please kindly refer to it. In addition, we added some qualitative results to show our effectiveness of splitting two consecutive facial expressions more intuitively.

---

> > ### Comment · Reviewer_nGC4 · 2024-11-27
> >
> > Thanks for the authors' feedback.
> >
> > The 'random selection' strategy is interesting. However, the results in the table or figures are not convincing that 'random' is the best. First, ‘random’ is not always the best in the table. Second, it is not clear why 'random' could be the best. Intuitively, why not use the accumulated values of the MaE and ME?
> >
> > When I say 'class' I originally mean the basic emotional classes (e.g., 'happy', 'sad') instead of MaE and ME. In other words, it would be better if there was a clear definition of 'emotional-class-agnostic' intensity.

---

> ### Author Response · Authors · 2024-11-19
> **Response to Reviewer nGC4 (part 2)**
>
> **Q2. The presentation of the inference lacks sufficient detail. Specifically, it would be helpful to clarify what the outer-inner-contrastive score is and how these scores contribute to determining the temporal range and position of the facial expressions.**
>
> **A2.** Please allow us to explain the process in detail. As described in section 3.6.3, once we obtain the intensity scores **a** and action scores $\mathbf{S}$, we use multiple thresholds for **a** to generate multiple candidate expression proposals. Each proposal includes consecutive frames with intensity scores higher than the threshold, allowing us to determine the temporal range (i.e., the onset and offset frames), and the expression type is determined by thresholding the action score of the pseudo-apex frame, which has the highest intensity score within each proposal. However, due to the use of the multi-threshold strategy, many overlapping proposals are generated; therefore, we calculate a confidence score for each proposal and discard redundant ones with lower confidence scores.
>
> The confidence score for each proposal is calculated based on the outer-inner-contrastive (OIC) score, which can be calculated as follows:
>
> Suppose we have a proposal whose onset frame and offset frame are $s_i$ and $e_i$.
>
> Inner score: $\mathrm{Score_{inner}} = \frac{1}{L} \sum_{t = s_i}^{e_i} a_t$
>
> Outer score: $\mathrm{Score_{outer}}=\frac{1}{\frac{L}{2}}\left(\sum_{t=s_i- \frac{L}{4}}^{s_i - 1} a_t + \sum_{t = e_i + 1}^{e_i + \frac{L}{4}} a_t \right)$
>
> $\mathrm{Score_{OIC}} = \mathrm{Score_{inner}} - \mathrm{Score_{outer}}$
>
> After obtaining the OIC scores of each proposal, we compare them and discard highly overlapping proposals with lower OIC scores.
>
> In the revised manuscript, we have added some of the details to Section 3.6.3, please kindly refer to it.
>
> **Q3. The soft pseudo-label is established based on the predicted intensity scores. However, these predicted intensity scores can fluctuate throughout the training process. It would be helpful to clarify how the soft pseudo-label is updated in response to these changing intensity scores.**
>
> **A3.** To ensure that the intensity branch accurately represents the expression intensity of each frame, we begin training with samples that can be assigned confident pseudo-labels. Therefore, we start with a small number of warm-up epochs, during which we conservatively assign confident pseudo-labels within a limited range to learn basic knowledge that high intensity output corresponds to high expression intensity, and vice versa.
>
> After warm-up epochs, we employ our proposed GIM to identify the pseudo-apex frame around each point-label and assign soft pseudo labels. During the training phase, we repeat this soft pseudo-labeling process at every epoch, updating the pseudo-apex frame and the range for assigning soft pseudo-labels based on updated features and intensity output, to enhance the intensity-related feature learning.
>
> We emphasized the process about ‘repeat this soft pseudo-labeling process at every epoch’ in Section 3.6.2 to clarify this part, please kindly refer to it.
>
>
> **Q4. The legend of Fig 4 is unclear. The marks of ground-truth boundary frame are missing.**
>
> **A4.** Thank you for pointing out this issue. In each subfigure of Figure 4, the two dotted green lines (now we change the color to orange) represent the ground-truth boundaries of the expression instance: the left line marks the ground-truth onset frame, and the right line marks the ground-truth offset frame. The line graph with dots in each subfigure represents the soft pseudo-labels we assigned, which includes information about the estimated duration and the pseudo-apex frame (indicated by the peak of the line graph). For example, in Fig. 4(a), the ground-truth boundaries are frame 1 and frame 13, and our estimated duration spans from frame 1 to frame 11.
>
> We have updated Fig. 4 to enhance its clarity in our revised manuscript, please kindly refer to it.

---

> ### Author Response · Authors · 2024-11-27
>
> Dear Reviewer nGC4, thank you for your response.
>
> **Q1. The 'random selection' strategy is interesting. However, the results in the table or figures are not convincing that 'random' is the best. First, ‘random’ is not always the best in the table. Second, it is not clear why 'random' could be the best. Intuitively, why not use the accumulated values of the MaE and ME?**
>
> **First, ‘random’ is not always the best in the table.**
>
> When reporting and comparing performance with other methods or in ablation studies, our first priority is the overall F1-score. The separate ‘MaE’ and ‘ME’ F1-scores are calculated while obtaining the optimal overall F1-score. Thus, comparing the overall F1-score can verify the performance of our method, as explained in Section 4.2, lines 464–466.
>
> In the table, the results indicate that the ‘random strategy’ achieves the optimal overall performance on two datasets.
>
> **Second, it is not clear why 'random' could be the best. Intuitively, why not use the accumulated values of the MaE and ME?**
>
> The table compares possible strategies for generating soft pseudo-labels for a single frame, considering multiple consecutive instances of the same class (all MaEs or all MEs). Accumulating all the soft pseudo-labels for a single frame as the intensity supervision signal presents challenges because the number of labels generated for a single frame is unpredictable, and the accumulated value could exceed 1.0, which cannot represent the expression intensity.
>
> We compared three possible strategies and found that ‘random selection’ performs the best, as it can prevent our model from being biased toward either high-intensity or low-intensity output.
>
>
> **Q2. When I say 'class' I originally mean the basic emotional classes (e.g., 'happy', 'sad') instead of MaE and ME. In other words, it would be better if there was a clear definition of 'emotional-class-agnostic' intensity.**
>
> As the initial step in facial expression analysis, facial expression spotting (FES) aims to localize all facial expression instances. This task does not consider the specific emotional classes of each MaE or ME.
>
> We emphasized the definition of "class-agnostic intensity" in the revised manuscript. Specifically, we clarified that the intensity scores and action scores we estimated are independent of emotional categories. Please kindly refer to lines 218-220 in Section 3.2.

---

> ### Comment · Reviewer_nGC4 · 2024-11-28
>
> Thanks for the authors' detailed answers.
>
> Although certain details of the paper do not fully convince me, I find the overall design of the task and the approach interesting. I am inclined to weakly accept this paper.

---

> > ### Author Response · Authors · 2024-11-29
> >
> > We sincerely thank you for your thorough reviews and for your appreciation of our work.

---

### Author Response · Authors · 2024-11-19
**General Response**

We sincerely thank all the reviewers for their constructive comments and for kindly appreciating our motivation and insights. We would like to address all the concerns raised by the reviewers. We separately respond to individual reviewers to address their concerns. Additionally, we revised the manuscript in response to their suggestions, with changes highlighted in red.

The main changes in the revised manuscript regarding the reviewers’ concerns can be summarized as:

1.	We revised Fig. 4 and included additional details to enhance its clarity;

2.	We added more detailed descriptions in Section 3 and 4 regarding the inference process, hyper-parameter setting, and specific ablation studies.

3.	We added an appendix for the further discussion about: 1) disccusion on consecutive facial expression spotting; 2) the choice of hyper-parameter $k_c$ for MaE; and 3) evaluation of pseudo-apex frame detection.

Other changes:

We corrected the range for the rough duration estimation in Eq. (3), which was a typographical error and does not affect any of the original methodology, experimental results, or conclusions. We are sorry for the typo.

---

### Meta-Review · Area_Chair_gsKU · 2024-12-21

**Metareview:**

Weak Accepted:
The rebuttal provided clarifications about the proposed method and its analysis that were useful for assessing the paper's contribution and responded adequately to most reviewer concerns. All reviewers recommend acceptance after discussion (with marginally above the acceptance threshold), and the ACs concur. The final version should include all reviewer comments, suggestions, and additional clarifications from the rebuttal.

**Additional Comments On Reviewer Discussion:**

NA

---

### Decision · Program_Chairs · 2025-01-22

Accept (Poster)